# iVar, an Interpretation-Oriented Tool to Manage the Update and Revision of Variant Annotation and Classification

**DOI:** 10.3390/genes12030384

**Published:** 2021-03-08

**Authors:** Sara Castellano, Federica Cestari, Giovanni Faglioni, Elena Tenedini, Marco Marino, Lucia Artuso, Rossella Manfredini, Mario Luppi, Tommaso Trenti, Enrico Tagliafico

**Affiliations:** 1Center for Genome Research, University of Modena and Reggio Emilia, 41125 Modena, Italy; sara.castellano@unimore.it (S.C.); federica.cestari@gmail.com (F.C.); 2Department of Medical and Surgical Sciences, University of Modena and Reggio Emilia, 41124 Modena, Italy; mario.luppi@unimore.it; 3PhD Program in Clinical and Experimental Medicine (CEM), University of Modena and Reggio Emilia, 41121 Modena, Italy; 4Nabla2 s.r.l., 41124 Modena, Italy; giovanni.faglioni@gmail.com; 5Department of Laboratory Medicine and Pathology, Diagnostic Hematology and Clinical Genomics Unit, Modena University Hospital, 41124 Modena, Italy; tanedini.elena@aou.mo.it (E.T.); marino.marco@aou.mo.it (M.M.); artuso.lucia@aou.mo.it (L.A.); t.trenti@ausl.mo.it (T.T.); 6Centre for Regenerative Medicine, Life Sciences Department, University of Modena and Reggio Emilia, 41125 Modena, Italy; rossella.manfredini@unimore.it

**Keywords:** next-generation sequencing, database, variant annotation, variant classification, data management, clinical genomics

## Abstract

The rapid evolution of Next Generation Sequencing in clinical settings, and the resulting challenge of variant reinterpretation given the constantly updated information, require robust data management systems and organized approaches. In this paper, we present iVar: a freely available and highly customizable tool with a user-friendly web interface. It represents a platform for the unified management of variants identified by different sequencing technologies. iVar accepts variant call format (VCF) files and text annotation files and elaborates them, optimizing data organization and avoiding redundancies. Updated annotations can be periodically re-uploaded and associated with variants as historically tracked attributes, i.e., modifications can be recorded whenever an updated value is imported, thus keeping track of all changes. Data can be visualized through variant-centered and sample-centered interfaces. A customizable search function can be exploited to periodically check if pathogenicity-related data of a variant has changed over time. Patient recontacting ensuing from variant reinterpretation is made easier by iVar through the effective identification of all patients present in the database carrying a specific variant. We tested iVar by uploading 4171 VCF files and 1463 annotation files, obtaining a database of 4166 samples and 22,569 unique variants. iVar has proven to be a useful tool with good performance in terms of collecting and managing data from a medium-throughput laboratory.

## 1. Introduction

Next Generation Sequencing in clinical contexts has rapidly evolved; automated wet lab procedures, as well as sequencing platforms and data analysis pipelines, have become increasingly reliable, producing progressively more genomic data. The ever-growing number of genetic variants related to disease, i.e., variant interpretation, has now become the major challenge for clinical genomics laboratories. Hence, advancing knowledge of genetic variants necessitates expanding databases for variant interpretation. Consequently, information regarding the pathogenicity of genomic variants, from biological and clinical data, increases dynamically with new data. The interpretation of pathogenicity may change over the time, which obliges clinical diagnostic laboratories to manage the reinterpretation of variants and to be aware of ethical issues. This is particularly relevant for variants classified as of uncertain or unknown significance (VUS). These variants, for which the current scientific knowledge does not allow classification as either pathogenic/likely pathogenic or benign/likely benign, are the most challenging for both the clinical and psychological management of patients. Changing the classification of a variant previously denoted as a VUS may represent a major issue, especially when genomics are used to diagnose hereditary diseases; it may also impact the management of the affected carriers and the choice to extend the testing to all the potential carriers within a family.

Multigene panels for hereditary cancer risk assessment have shown an overall VUS range of 34–41% [1,2]. In particular, it has been reported that 7.3% of patients who underwent Next Generation Sequencing (NGS) testing with a hereditary cancer multigene panel were diagnosed with variants that were subsequently reclassified, and that 94% of the reclassified variants caused a change in these patients’ clinical management [3].

The issue of reinterpretation of variants in molecular genetics laboratories is leading to the need for guidelines and tools [4,5]. Patient care would likely benefit from the reclassification of variants through the use of robust data management systems and organized approaches to variant reinterpretation [6].

Currently, there is no consensus on how a clinical laboratory should revise the classification of variants in patients tested in the past, nor on how frequently this should be done. However, despite the absence of a clear legal duty to recontact patients after revision of genomic test results, the ethical responsibility of the clinical laboratory to inform clinicians about variant reclassification has been considered. Pursuing this goal involves optimization of the limited available resources [6,7,8]. Appelbaum and collaborators recently conceptualized the ethical duty to reinterpret genetic variants by identifying four elements: data storage, initiation of reinterpretation, data reinterpretation and recontacting patients [6]. Accessibility to data is an obvious basic prerequisite to reinterpretation. Reinterpretation can be periodical, when a certain number of changes in interpretation are accumulated, or can be a stakeholder decision. Data reinterpretation involves both a data analytic pipeline and human judgment. Recontacting patients is the responsibility of clinicians once they have the reinterpreted results.

Laboratories should therefore be able to identify previous results from patients with particular variants in order to start recontacting. Suitable informatics tools can simplify the process, making recontacting as efficient as possible.

Herein we present iVar, a freely available tool with a user-friendly web interface. This tool can help to fulfil the aforementioned duties by providing a platform for the unified management of variants identified by different sequencing technologies. Most importantly, iVar is a useful tool for easily setting up an automated process of periodic re-annotation of variants that allows users to check if the pathogenicity of a variant has changed. Furthermore, recontacting patients is made easier through the effective identification of all patients present in the database who carry a specific re-annotated variant. In addition, a high level of customization is provided, potentially allowing the uploading of all the VCF files generated by different tools, including free (e.g., GATK or samtools) and commercial software (e.g., Torrent Suite Software or MiSeq Reporter). Additionally, the annotation files arising from any bioinformatic pipeline can be uploaded. Because the database can easily be queried by users even without bioinformatics expertise, it can work as a valuable tool to assist geneticists and clinicians in retrieving data for statistical analyses.

## 2. Materials and Methods

### 2.1. Implementation of iVar Workflow

iVar package is publicly available at the GitHub repository (https://github.com/CGR-UNIMORE/iVar (accessed on 12 February 2021)).

iVar takes VCF files and annotation files as input (Figure 1). 

Variant Call Format (VCF) is a text file format containing meta-information lines, a header line and data lines, each containing information about a position in the genome. A VCF file also has the ability to contain genotype information about samples for each position. Because different variant caller software can generate slightly different VCF files, users must define the specific format of the VCF files to be imported through a simple web interface. Moreover, a predefined “gene panel” has to be associated with each imported VCF file to indicate the genes included. Additionally, a “virtual panel” filter can be set up for VCF files to import only data lines containing variants included in a predefined gene list. This ensures that only gene variant data complying with informed consent rules are imported if extended gene panels are utilized for sequencing. The second type of file that can be imported is the annotation file, which is a text file containing data obtained from custom annotation pipelines or commercial tools for variant annotation and classification. As with files in the VCF format, the import format for an annotation file can be customized.

Imported and structured data can be viewed through different interfaces. In particular, users can examine a list of variants to evaluate the pathogenicity class label by accessing the annotation information, which relies on the previously defined annotation file and the associated variants found in tested samples. Additionally, it is possible to generate a list of samples and check their variants for pathogenicity classification, allele frequency and genotype. 

Variant annotations can be kept up-to-date through a process that we term “re-annotation”. This consists of three main steps: (i) export of variants from iVar in VCF format; (ii) annotation of the exported variants outside iVar, through custom annotation pipelines or other existing annotation tools; and (iii) import of the new annotation file back into iVar. Annotation values are tracked historically, i.e., modifications are recorded whenever an updated value is imported, thus keeping track of all changes over time.

To assess if relevant information changes upon re-annotation or at a particular time, a customizable search function is provided. With this tool, users can specify search conditions for attributes of interest and can assess changes to these attributes (e.g., changes in the ClinVar attribute from “benign” or “uncertain significance” to “pathogenic”).

### 2.2. Software and Hardware Implementation

The iVar database was developed under Ubuntu 18.04 LTS Linux operating system 64 bit (a 32-bit Linux has a maximum database table limit of 2 GB, which is too small). The software was implemented using Python (version 2.7), Web2py Framework (version 2.18.5), Bootstrap4 toolkit and MariaDB (version 10.3.18) SQL Database backend. Apache (version 2.4.29) and phpMyAdmin (version 4.6.6-5) were used as development tools. iVar was built as a platform for collaborative developing with a responsive interface for both PC and mobile applications.

For database development and testing, a workstation with the following hardware specifications was used: Intel Xeon CPU E3-1231 v3 3.40 GHz; 8 GB RAM; 1 TB disk. For security, all HTTPS were created with a Let’s Encrypt Authority X3 certificate; MariaDB Data-at-Rest Encryption for back-up and HDD disposal safety; and FS Encryption for the VCF files.

## 3. Results

In order to test all the iVar features, 4171 VCF files (from different analysis platforms) and 1463 annotation files (produced both by our custom annotation pipeline and by SOPHiA DDM annotation pipeline) were uploaded to populate the attributes of variants. During the uploading step, VCF files were filtered with patients’ informed consent and, where appropriate, a virtual gene panel filter was added. A total of 14 gene panels and 44 virtual gene panels, comprising 301 genes, were set up to filter variants associated with the following clinical pathologies: breast and ovarian hereditary cancer, dyslipidemic disorders, epidermolysis bullosa, hemochromatosis, nephropathies and retinitis pigmentosa. 

### 3.1. VCF Files Upload

For testing purposes, VCF files generated both by the Torrent Suite Software (TSS) (Saxena et al.) (VCF version 4.1) and by SOPHiA DDM (DDM) (VCF version 4.2) were used. 

The TSS VCF files were generated using a custom hotspot file containing 5425 variants, including pathogenic variants from ClinVar, Enigma and LOVD. Each TSS VCF file included a large number of variants, resulting from the hotspot file, where allele frequency was 0 and genotype was 0/0. Hence, a row filter excluding all the lines where the subfield genotype (GT) is 0/0 was set when uploading TSS VCF files. Also, we defined the subfields allele frequency (AF) and genotype (GT), included in the VCF file “FORMAT”, as the attributes linked to the sample, when importing the VCF file. This showed the allelic frequency and the genotype of the variants found within each sample.

For the DDM-generated VCF files, no filters were applied because no AF subfield was present. Therefore, to make those files congruent with the sample attributes defined in the TSS-generated VCF files, we set up the SOPHiA DDM VCF file type by exploiting the allelic depth (AD) and read depth (DP) of the subfield. The resulting sample attribute is defined as: (1)$$\frac{AD}{DP}\times 100$$

Furthermore, considering that DDM VCF files contain additional information in the “INFO” field, we took advantage of some subfields to link these attributes to the variant when importing the VCF file. Specifically, we obtained the gene symbol of the variant from the “SGVEP” subfield, information from ClinVar and other databases from the “DBXREF” subfield, and mutation type from the “TYPE” subfield.

After defining the VCF file types, 2798 TSS VCF files and 1373 DDM VCF files were uploaded. Next, the number of variants for each imported sample was verified and random checks were performed on variant and sample attributes, paying particular attention to variants present in both VCF file types. All VCF files were correctly elaborated by iVar and, importantly, we ascertained that when the same variant was imported from two different VCF file types, the common attributes were overwritten if identical, or tracked historically if different, thereby preventing redundancies.

### 3.2. Annotation Files Upload (Text Files Upload)

Annotation files were uploaded after the VCF files. In particular, using the iVar annotation files definition tool, nine different variants annotation types were defined:AT1: four annotation types for files generated by our customized annotation pipelines, which annotate Ion Torrent VCF files interrogating databases related to four different pathologies: breast and ovarian cancer, dyslipidemias, epidermolysis bullosa and hemochromatosis;AT2: three annotation types for files generated by our customized annotation pipelines, which annotate SOPHiA DDM VCF files interrogating databases related to three different pathologies: breast and ovarian cancer, dyslipidemias and nephropathies;AT3: one annotation type for files generated by the SOPHiA DDM software; andAT4: one annotation type for files coming from periodical re-annotation with our custom pipeline of VCF files exported from iVar and containing unique variants.

Different annotation types may have common attributes; therefore, we used the same names for common attributes when defining the annotation type setting. This prevented redundancies and allowed tracked historical values to be applied when they differed. Additionally, row filters and break conditions could be set if the annotation files contained lines to be skipped, or if only a part of the file was to be imported.

A total of 54 AT1 annotation files were imported; of these, 45 were breast and ovarian cancer, 3 were dyslipidemias, 5 were epidermolysis bullosa and 1 was hemochromatosis. A total of 61 AT2 annotation files were imported; of these, 48 were breast and ovarian cancer, 7 were dyslipidemias and 6 were nephropathies. There were 1344 AT3 annotation files and 4 AT3 annotation files.

Random checks performed on variant attributes showed that all the annotation files were correctly imported and elaborated in iVar.

### 3.3. VCF and Annotation Uploading Performance Testing

We tested VCF file uploading and elaboration performance for both types of VCF files. A typical TSS VCF file (5000 rows, 3.2 MB) was imported in 1 s and was elaborated in 5 s. Due to the row filter described above, only 15 variants of 5000 were uploaded on average. A typical SOPHiA DDM VCF file (500 rows, 150 KB) also took 1 s to be imported, but the elaboration time depended largely on the applied virtual panel type. Specifically, it took, on average, 4 s to elaborate a VCF file with a two-gene virtual panel (50 uploaded variants on average), 6 s for a VCF file with a five-gene virtual panel (70 uploaded variants on average) and 21 s for a VCF file with a 22-gene virtual panel (400 uploaded variants on average).

These results indicated that for larger files containing over 5000 variants, but associated with a very strict row filter, processing times were short. Elaboration times were slightly longer for a smaller VCF file when more variants were uploaded. With our default innodb_page_size of 16KB, our maximum tablespace size was 64 TB, which was greater than our hard disk size. In one particular case, in order to import a 24MB VCF file, we had to increase max_allowed_packet to 1 GB and key buffer to 64 MB. The limitations of iVar, e.g., limitations on schema, size, tables, and transactions and locks, were inherited from MariaDB (https://mariadb.com/kb/en/innodb-limitations/ (accessed on 9 November 2020)).

For all the annotation file types, uploading took 1 s per file. Elaboration times varied, depending mainly on the number of attributes in the annotation file type, the number of variants and the number of nonempty attributes in the annotation file. We observed that it took 25 s on average to import an AT1 annotation file (with varying numbers of annotations for 90 variants), 2 min for an AT2 annotation file (with varying numbers of annotations for 500 variants) and 1 min for an AT3 annotation file (with varying numbers of annotations for 500 variants). Differences between elaboration times for AT2 and AT3 annotation files, which included the same average number of variants, were due to the AT3 annotation type containing about half as many attributes as the AT2 type.

As a result, we obtained a database containing 4166 samples and 22,569 unique variants with about 283,659 annotation attributes and, among them, 22,501 variants associated with at least one attribute (Table 1). The total size of all of the iVar database tables was about 1.2 GB.

Using the five-tier International Agency for Research on Cancer (IARC) classification system [9], and following the guidelines of the American College of Medical Genetics and Genomics (ACMG) for the interpretation of sequence variants [10], we assigned a pathogenicity classification to 1016 variants (Table 2). Classification was batch uploaded using a tab-delimited text file, for which an appropriate text file type was previously set. Variant classification, however, was either imported along with the other variant attributes within an annotation file or set directly in iVar via the web interface.

### 3.4. Queries Functionality Testing

A set of common queries was executed for performance testing. A simple variant search of 22,569 variants was completed in 1 s, while searching for all variants classified as C5 (330 resulting variants) required 2.3 s. Searches for variants and their attributes took, on average, 2 s.

### 3.5. iVar Variants Export

The “export vcf for re-annotation” functionality allows users to export all the unique variants present in the database as a VCF file, which is readily parsable with any annotation pipeline. We timed this feature by exporting the variants after data upload. It took 1 s to export a ZIP file containing three VCF files, with up to 10,000 variants in each.

### 3.6. Re-Annotation

To test the re-annotation pipeline for all variants in the database, we exported the 22,569 unique annotated variants in the iVar database, annotated them using our pipeline and re-imported the resulting file back into iVar. It took 3 s to import the file (AT4 annotation file type) into iVar and 30 min to elaborate it.

### 3.7. Functionality for Annotation Changes Check

The “variants and attributes search” function can assess whether some attribute values changed after re-annotation. This is a customizable search functionality that allows users to set different types or searches to answer clinical questions. The functionality allows search conditions for variants, attributes and previous attributes. To time it, we defined search criteria to identify the variants in which ClinVar values changed from “benign” or “uncertain significance” to “pathogenic”. This search took about 3 sec and produced 11 results. This search also allowed the identification of a BRCA2 variant (chr13|32936829|A|G, NM_000059.3:c.7975A>G) that was consequently reclassified from C3 (uncertain significance) to C5 (pathogenic), which allowed us to enroll a new patient in the surveillance program.

## 4. Discussion and Conclusions

iVar has proven to be a useful platform for collecting and managing data from our diagnostic laboratory, where genomic data related to hereditary diseases are regularly produced using different sequencing technologies. This first version of the software was designed as a proof of concept, neglecting possible optimizations of the database structure, yet still showing good performance in the management of our medium-throughput laboratory. In particular, we uploaded 4171 VCF files produced by our laboratory since 2014, using 44 different virtual panels. We also tested our tool handling of a volume of data about 10 times larger, with acceptable performance. The management of massively larger amounts of data is very likely to require further database optimization, e.g., of the sizes of caches, buffers or other parameters. In some instances, it could require more powerful hardware.

The iVar structure was designed to be highly customizable, and it is particularly suited to handle information from different types of input files in order to optimize data organization and prevent redundancies. We uploaded VCF files produced by two different technologies and 1463 annotation files produced by different annotation pipelines, using both custom and commercially provided software.

The functionality for evaluating annotation changes is a powerful tool to manage reinterpretation of variants in molecular genetics laboratories. Exploiting this functionality, we were able to identify a BRCA2 variant, which was previously classified as “uncertain significance”, and to reclassify it as a “pathogenic” variant on the basis of the updated ClinVar annotation.

In the current version, this process is performed manually after each re-annotation, or whenever changes occurring after a certain date are to be evaluated. In the future, we plan to implement an automatic re-annotation process, to be run in the background periodically. This would enable notification to the users whenever significant changes in variant annotations occur. Users will then verify the extent of these changes to assess possible variant reclassifications, identify patients and start the recontacting process.

iVar could be further improved by linking the database to an existing database containing clinical data, using a unique identifier for each patient. This would allow the integration of clinical and genomic data, and consequently, make iVar a useful tool with which to investigate family relationships and study genotype–phenotype correlations.

## Figures and Tables

**Figure 1 genes-12-00384-f001:**
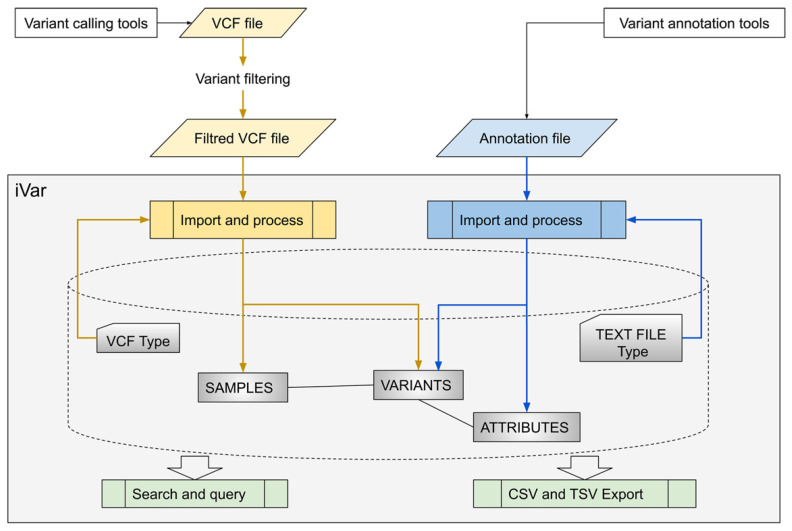
Overview of iVar, a platform for the unified management of variants identified by different sequencing technologies. CSV Comma Separated Value; TSV Tab Separated Value; VCF Variant Call Format.

**Table 1 genes-12-00384-t001:** Description of regions and types of variants uploaded in iVar.

	Number of Variants with at Least 1 Attribute
Exonic regions	6084
Exonic splicing junctions	3
Exonic regions of noncoding transcripts	52
Intronic regions	12,801
Intronic regions of noncoding transcripts	524
Splicing junctions	90
3’-UTR	1648
5’-UTR	415
Missense	5026
Nonsense	63
Synonymous SNV	3169
Variants affecting splicing	720
Frameshift	711

SNV—single nucleotide variant.

**Table 2 genes-12-00384-t002:** Summary of classified variants included in iVar.

Pathogenicity Classification of 1100 Annotated Variants	Total Number (%)
C5 (Pathogenic)	330 (32.5%)
C4 (Likely Pathogenic)	38 (3.7%)
C3 (Uncertain Significance)	333 (32.8%)
C2 (Likely Benign)	206 (20.3%)
C1 (Benign)	109 (10.7%)

## Data Availability

iVar package is publicly available at GitHub repository (https://github.com/CGR-UNIMORE/iVar).

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
