# Peer review of "iVar, an Interpretation-Oriented Tool to Manage the Update and Revision of Variant Annotation and Classification"

_genes, 2021, doi:10.3390/genes12030384_

Round 1

Reviewer 1 Report

Dear Authors

your manuscript appears at the right time. Variant annotation tools already exist and several of them are of very sufficient quality for the interpretation of genetic data. However, your approach, which you call "reannotation", is particularly interesting. Indeed, more and more articles are being published presenting reinterpretations of data both for cancer and genetic diseases, reannotation are indispensable in the follow-up of patients. I am therefore very enthusiastic about the use of your tool.

The only drawback I detect, which may be an inconvenience, is the visualization of the coverage that does not appear clearly in your tool. Tools exist for that, it would be good to be able to implement them to monitor whether the targeted areas are deeply sequenced or not.

L220 please modify it takes took ...

Author Response

We are very happy that the reviewer enjoyed this work and is enthusiastic about our tool. He rightly proposed adding a coverage visualization to monitor the sequencing depth of the targeted areas. We agree that this is important information to be aware of. Unfortunately, a coverage visualization of entire targeted areas would not be feasible in this case, as iVar accepts VCF files as input and by its nature, VCF file does not contain such information. Usually, a coverage analysis is performed starting from aligned bam files. Sometimes, however, depending on the variant caller that generated the VCF file, information on the coverage of a particular genomic position may be present (usually it can be found in the “DP” field). In this case, the VCF file type in iVar can be properly defined to get this information, which will be visualized as an attribute linked to the sample. In the same way, as described in section 3.1 of the manuscript (L168 - L171), we obtained information on allelic frequency (AF) and genotype (GT) from VCF file. Precisely because VCF data is little standardized, we decided to keep the import and visualization of information coming from VCF files as customizable as possible. For this reason, we did not implement a predefined section for coverage visualization in iVar.

In addition to the above comments, the error on L220, pointed out by the reviewer, has been corrected.

Reviewer 2 Report

(1) I think the tool is only useful for the clinical labs that use NGS for screening the patients. The tool is not very useful for researchers. Therefore, the significance of this manuscript is not high.

(2) The tool is not a standalone tool and requires many other tools to be installed first. The installation of the is not trivial so I think the current form will limit its applicability.

On the other hand, the paper is well written and the tool is well described. I have no objection if you think the tool is still useful and the manuscript is publishable at the current format. 

Author Response

We agree with the referee that this tool has been mainly designed for diagnostic laboratories when a periodic re-annotation of variants is required for an eventual re-contact of patients for whom this could be important. Despite this, this tool could also be useful to researchers who do clinical research and genotype-phenotype association studies when variants considered not important in determining a phenotype may assume a different pathological value.

Regarding the easier handling of the software. We definitely agree with their comments. Therefore, we implemented, and uploaded to the Github repository (https://github.com/CGR-UNIMORE/iVar), a fully automated installation script, tested on Ubuntu 18.04 lite server. We are also considering a docker container solution, as we understand it would be more amenable for non-technical users.

Reviewer 3 Report

Castellano et al. present iVar, an interpretation and annotation tool for next-generation sequencing data. The authors present a case for the necessity of being able to easily re-annotate and re-interpret clinical variant data as annotations improve and change over time.

Castellano et al. have described the construction of an open source (thumbs up!) and freely available bioinformatics tool to annotate and help interpret variant call files (VCFs) and go on to present workings of the software on their gene panel sequencing data. This is a well described tool, clearly illustrated as to its functionality and all source code has been made available on GitHub. iVar should be of great use to the wider clinical bioinformatics sequencing field.

Have the authors considered the performance of the tool applied to whole exome, or even whole genome sequencing data? Much of the genetic testing is now moving to whole genome sequencing.

Author Response

The reviewer, interestingly, asks if we considered the performance of the tool applied to whole exome or whole genome sequencing data. We designed and adjusted this first version of the software on targeted sequencing data from a medium throughput diagnostic laboratory like ours, neglecting possible optimizations of the database structure. We considered that whole exome and, in particular, whole genome sequencing, are currently exploited mainly in the research field. However, we are aware that these kinds of analyses could become more and more widespread in the future, even in the diagnostic field. It would therefore be interesting to adapt the software to achieve good performance even with much higher data throughput. For this reason, as mentioned in the manuscript (L228), we tested the import of a 24MiB VCF file, which was generated by a whole exome sequencing experiment, and we also tested our tool handling a volume of data about 10 times larger than that considered in this work, obtaining acceptable performances. Nevertheless, we recognize that, as already described in the discussion section of the manuscript, the management of massively larger amounts of data, is very likely to require further database optimizations, such as the sizes of caches, and buffers, or other parameters. In some instances, it could require more powerful hardware.

Round 2

Reviewer 2 Report

The authors have responsively addressed the comments from three reviewers. I am happy to see that the authors provided a fully automated installation script to make the use of iVar much easier. I do not have any further comments.